# Multi-model ensembles for assessment of flood losses and associated uncertainty

Rui Figueiredo[1,2], Kai Schröter[2], Alexander Weiss-Motz[2], Mario L.V. Martina[1], Heidi Kreibich[2]

[1] Scuola Universitaria Superiore IUSS Pavia, Pavia, Italy

[2] GFZ German Research Centre for Geosciences, Section 5.4: Hydrology, Potsdam, Germany

*Correspondence to*: Rui Figueiredo (rui.figueiredo@iusspavia.it)

**Abstract.** Flood loss modelling is a crucial part of risk assessments. However, it is subject to large uncertainty that is often neglected. Most models available in the literature are deterministic, providing only single point estimates of flood loss, and large disparities tend to exist among them. Adopting any one such model in a risk assessment context is likely to lead to
inaccurate loss estimates and sub-optimal decision-making. In this paper, we propose the use of multi-model ensembles to address these issues. This approach, which has been applied successfully in other scientific fields, is based on the combination of different model outputs with the aim of improving the skill and usefulness of predictions. We first propose a model rating framework to support ensemble construction, based on a probability tree of model properties, which establishes relative degrees of belief between candidate models. Using twenty flood loss models in two test cases, we then construct
numerous multi-model ensembles, based both on the rating framework and on a stochastic method, differing in terms of participating members, ensemble size and model weights. We evaluate the performance of ensemble means, as well as their probabilistic skill and reliability. Our results demonstrate that well-designed multi-model ensembles represent a pragmatic approach to consistently obtain more accurate flood loss estimates and reliable probability distributions of model uncertainty.

## 1. Introduction

Effective management of flood risk requires comprehensive risk assessment studies that consider not only the hazard component, but also the impacts that the phenomena may have on the built environment, economy and society (Messner and Meyer, 2006). This integrated approach has gained importance over recent decades, and with it so has the scientific attention given to flood vulnerability models describing the relationships between flood intensity metrics and damage to physical
assets, also known as flood loss models. A large number of models have become available in the scientific literature. However, despite progress in this field, many challenges persist in their development, and flood loss models tend to be quite heterogeneous. This often results in practical difficulties when they are to be applied in risk assessment studies (Gerl et al., 2016; Jongman et al., 2012), as described below.

Flood damage mechanisms are complex, being dependent on different properties of flood events, such as water depth, flow
velocity and flood duration, as well as on the physical characteristics of the exposed assets (Kelman and Spence, 2004).

Precautionary and socio-economic factors can also influence their degree of vulnerability (Thieken et al., 2005). Building accurate and reliable flood loss models that account for all these factors is a challenging task. Model development is hampered by limited knowledge about damage-influencing factors, as well as limited data availability (Merz et al., 2010). It is therefore unsurprising that traditional flood loss models tend to be rather simple, often using water depth as the only

explanatory variable to describe damage and loss to coarsely defined groups of assets (Green et al., 2011; Smith, 1994). However, the limited predictive ability and high degree of uncertainty associated with such models has been acknowledged (Krzysztofowicz and Davis, 1983; Merz et al., 2004), and more complex models that consider additional explanatory variables have been developed (Dottori et al., 2016; Elmer et al., 2010; Merz et al., 2013). Regardless, uncertainty in flood loss modelling is to some extent inevitable (Schröter et al., 2014).

Furthermore, flood loss models are usually developed for specific regions, ranging from country to catchment or municipality level, with smaller scales making up the majority of models (Gerl et al., 2016). Lack of available flood loss models in many regions often leads to the transfer of models in space, resulting in their application to contexts with different built environments and/or socio-economic settings than originally intended. However, this is generally done with insufficient justification, and flood loss models have been shown to offer lower predictive ability under such circumstances (Cammerer

et al., 2013; Jongman et al., 2012; Schröter et al., 2014).

In addition, flood loss models are most often constructed for specific flood types (e.g. fluvial flood, flash flood, coastal flood), and will usually be poorly suited to estimate loss due to flood events with other dominant damaging processes (Kreibich and Dimitrova, 2010; Kreibich and Thieken, 2008). Models also vary in the way loss is expressed, which can be either in monetary terms or as a fraction of the value of the element at risk (Messner et al., 2007). These are referred to

respectively as absolute and relative flood loss models, the latter being better suited than the former for application across different study cases (Krzysztofowicz and Davis, 1983). Further differences may exist in terms of other model attributes.

Due to this large heterogeneity, it is difficult to identify flood loss models that, given their attributes, are potentially the most appropriate for application in specific risk assessment studies. Ideally, for any given application setting, a perfectly suited model (e.g. similar type of asset, no spatial transferability required, validated with local evidence) would be available and

unambiguously identifiable, but unfortunately, this is far from the case. The lack of an established procedure to select suitable flood loss models from the many available in the literature means that model selection is often done rather arbitrarily (Scorzini and Frank, 2015), which can negatively impact the quality of flood loss estimations and lead to suboptimal investment decisions based on model outcomes (Wagenaar et al., 2016).

A critical issue in flood loss modelling is uncertainty (Merz et al., 2004), which is usually high and can significantly

contribute to overall uncertainty in flood risk analyses (de Moel and Aerts, 2011). Model uncertainty is mainly related with parameter representation, whereby fewer parameter than those theoretically needed to describe physical damage processes are used, and with insufficient data and/or knowledge about damage processes (Wagenaar et al., 2016). Quantifying uncertainty is imperative, as this information is required to make informed decisions in the context of flood risk management (Downton et al., 2005; Peterman and Anderson, 1999; USACE, 1992). However, the vast majority of flood loss models

currently available in the literature are deterministic (Gerl et al., 2016), providing single point estimates of loss. Such estimates are unable to meet the decision needs of different stakeholders, who may have differing risk attitudes or cost-benefit ratios for risk mitigation measures (Merz and Thieken, 2009). Moreover, the uncertain nature of flood loss estimations means that the performance of any given deterministic model that appears appropriate for a certain application

can be limited, as large disparities may exist even among seemingly comparable models (Jongman et al., 2012; Merz and Thieken, 2009). This makes flood risk estimates highly sensitive to loss model selection (Apel et al., 2009; Wagenaar et al., 2016). It is thus clear that adopting a single deterministic model for the estimation of flood losses is not recommended, as the information it provides is insufficient for optimal decision-making, and the results will potentially, and very likely, be inaccurate. Even though research on flood loss modelling has recently started to move into the probabilistic domain (Custer

and Nishijima, 2015; Dottori et al., 2016; Kreibich et al., 2017; Schröter et al., 2014; Vogel et al., 2012), probabilistic models are still scarce.

Multi-model ensembles have been successfully applied in scientific fields such as hydrology or weather forecasting to tackle similar issues to those discussed above. Ensemble means have been shown to almost always outperform individual models (Georgakakos et al., 2004; Gleckler et al., 2008; Reichler and Kim, 2008), and the combination of the output of different

models can be a pragmatic approach to estimate model uncertainty (Palmer et al., 2004; Weigel et al., 2008). However, in the context of vulnerability modelling, the concept of combining multiple models is relatively new. Rossetto et al. (2014) and Spillatura et al. (2014) have proposed the use of mean model estimates as part of their studies on respectively fragility and vulnerability curves for seismic risk assessment, but model performance is not evaluated and uncertainty quantification is not discussed. The potential use of multi-model ensembles in flood vulnerability assessment has not been addressed before.

This study therefore aims to answer the following research questions:

1. Can multi-model ensembles be used to improve the accuracy of flood loss estimations?
2. Are multi-model ensembles able to represent model uncertainty and provide reliable probabilistic estimates of flood loss?
3. How should such ensembles be constructed?

We first propose a framework to rate flood loss models according to their potential skill and suitability as participating members in such ensembles. We then construct various multi-model ensembles, based both on the rating framework and on a state of simulated non-informativeness, differing in terms of participating members, ensemble size, and weighting criteria, and evaluate their performance. Twenty flood loss models available in the literature are adopted, and losses are modelled for residential buildings in two application cases, corresponding to flood events that took place in Germany in 2002 and in Italy

in 2010. Based on the results, which are shown and discussed in Section 3, conclusions are drawn regarding the application of multi-model ensembles in flood loss estimations.

## 2. Setup of validation exercise

### 2.1. Flood loss models

The flood loss model catalogue developed by Gerl et al. (2016) was used as the basis for model selection in this study. We first identified all deterministic models describing loss to residential buildings, and then excluded models based on following criteria:

- The documentation is insufficient for model implementation;
- The model uses explanatory variables that are not available in most practical applications;
- The model has a functional form that is considered inappropriate (e.g. too simplistic or discretised);
- The model is based on the same dataset as another model deemed more appropriate for the application settings (this is to ensure model independence and avoid potential biases in the resulting ensembles).

Based on this procedure, twenty deterministic flood loss models for residential buildings were adopted. The catalogue developed by Gerl et al. (2016) provides information on the properties of each model, which is necessary to assess model suitability according to the framework proposed in Section 3.1. Table I shows the model properties relevant for this study, as well as the corresponding references, where model formulations can be consulted.

Each model is implemented to compute flood losses for the two application cases described in Section 2.3, for which the available hazard and exposure data are shown in Table II. This consists in the largest to date application of different flood loss models within the scope of a scientific study on flood risk. In the estimation of losses for each asset, the best-matching function from each model is selected. In cases where this cannot be done unambiguously (e.g. due to mismatch in asset description between the exposure dataset and the model documentation), the selection is based on expert judgement. When models do not use some of the available hazard or exposure data, the unused variables are not considered. Losses given in absolute terms are adjusted for inflation. The modelled losses are provided as supplementary material.

### 2.2. Evaluation methods

#### 2.2.1. Deterministic predictions

The predictive performance of single loss models and ensemble means is evaluated in terms of accuracy and systematic bias, using respectively the root mean squared error (RMSE) and the mean bias error (MBE). These are given by

$$RMSE = \sqrt{\frac{1}{n}\sum_{i=1}^{n}(\hat{X}_i - X_i)^2} \tag{2.1}$$

and

$$MBE = \frac{1}{n}\sum_{i=1}^{n}(\hat{X}_i - X_i),\tag{2.2}$$

where $\hat{X}$ is a vector of $n$ predictions and $X$ is the vector of observed values of flood loss.

### 2.2.2. Ensemble predictions

The probabilistic skill of ensembles is evaluated using the continuous ranked probability score (CRPS), which is defined as the integrated squared difference between the cumulative distributions of predictions and observations (Weigel, 2012). We adopt the expression for the CRPS derived by Hersbach (2000), which is described as follows. Consider a set of $n$ assets affected by a flood with corresponding observed losses $x_1, \dots, x_n$. Let there be $m$ ensemble members, and let $\hat{x}_{t,i}$ be the prediction of loss given by $i^{th}$ ensemble member for the $t^{th}$ asset, sorted in ascending order. Define $\hat{x}_{t,0} = -\infty$ and $\hat{x}_{t,m+1} = +\infty$. The CRPS is given by

$$CRPS = \frac{1}{n}\sum_{t=1}^{n}\left[\sum_{i=1}^{m}\alpha_{t,i}\left(\frac{i}{m}\right)^2 + \sum_{i=0}^{m-1}\beta_{t,i}\left(1 - \frac{i}{m}\right)^2\right]\tag{2.3}$$

where

$$\alpha_{t,i} = \begin{cases} 0 & if \quad x_t \leq \hat{x}_{t,i} \\ x_t - \hat{x}_{t,i} & if \quad \hat{x}_{t,i} < x_t \leq \hat{x}_{t,i+1} \\ \hat{x}_{t,i+1} - \hat{x}_{t,i} & if \quad \hat{x}_{t,i+1} < x_t \end{cases}$$

and

$$\beta_{t,i} = \begin{cases} \hat{x}_{t,i+1} - \hat{x}_{t,i} & if \quad x_t \leq \hat{x}_{t,i} \\ \hat{x}_{t,i+1} - x_t & if \quad \hat{x}_{t,i} < x_t \leq \hat{x}_{t,i+1}. \\ 0 & if \quad \hat{x}_{t,i+1} < x_t \end{cases}$$

The CRPS can be interpreted as an error measure, with lower values corresponding to higher probabilistic skill.

To assess ensemble reliability (i.e. whether ensemble predictions and observations are statistically indistinguishable), the rank histogram is adopted, which is constructed as follows. Consider an $m$-member ensemble prediction $\hat{x} = (\hat{x}_1, \dots, \hat{x}_m)$ and a corresponding observation $x$. The rank of $x$ in relation to the ensemble members of $\hat{x}$ is given by $r = M + 1$, where $M$ is the number of ensemble members that $x$ exceeds ($M \leq m$). For example, if $x$ is smaller than all ensemble members, the observation has rank $r = 1$, while if $x$ exceeds all ensemble members, then $r = m + 1$. If an ensemble is reliable, for a set of $n$ prediction-observation pairs there should be $n/(m + 1)$ observations with each $m + 1$ possible rank values, i.e., the histogram should be flat. Systematic deviations from flatness can indicate deficiencies in terms of ensemble dispersion and bias. Note that no ensemble is perfectly reliable, and random deviations from flatness are expected due to sampling uncertainty (Talagrand et al., 1998; Weigel, 2012).

### 2.3. Application cases

#### 2.3.1. 2002 flood along the Mulde River, Germany

Floods are a recurring natural hazard in the Mulde catchment (7 400 km²) located in Saxony, Germany. In recent years, this area has been severely affected by the June 2013 and August 2002 floods (Engel, 2004; Merz et al., 2014). The latter was triggered by record-breaking precipitation amounts in the Ore Mountains, which form the headwaters of the Mulde River. At the Zinnwald-Georgenfeld station, operated by the German Weather Service, 312 mm of rainfall were recorded within 24h (Ulbrich et al., 2003). The flood caused many dike breaches and resulted in considerable loss in 19 Saxonian municipalities along the Mulde (Figure 1).

The data used for this application case are listed in Table II, and the results of individual model applications in terms of error statistics are shown in Table III . The flood extension and water depths were estimated through hydro-numeric simulations (Apel et al., 2009) and hydraulic transformation (Grabbert, 2006). Return periods of flood peak discharges were derived from annual maximum series of mean daily discharges by Elmer et al. (2010). For the estimation of contamination indicators, inundation durations, flow velocity indicators and precautionary measures indicators, computer aided telephone interviews with affected households have been used (Thieken et al., 2005). The average floor areas of residential buildings and average building values are based on official statistical data about total living area for different types of residential buildings per district, and standard construction costs per square meter gross floor area (Kleist et al., 2006). Asset values with a spatial resolution corresponding to the inundation map (i.e. 10x10 m²) have been derived by applying a binary disaggregation method and using the digital basic landscape model ATKIS as ancillary information (Wünsch et al., 2009). Residential building type composition and mean residential building quality per municipality were derived by Thieken et al. (2008) using geo-marketing data from INFAS GEOdaten GmbH from 2001. Flood losses to residential buildings have been documented by the Saxon Relief Bank on the municipality level (Saxon Relief Bank 2005) and amount to a total of 240.6 million Euro. For more details, see Kreibich et al. (2017).

#### 2.3.2. 2010 flood in Caldogno, Italy

From 31 October to 2 November 2010, the Veneto Region was affected by persistent rain, particularly in the pre-Alpine and foothill areas, with accumulated rainfall exceeding 500 mm in some locations (Regione del Veneto, 2011a). This caused multiple rivers to overflow, resulting in floods that inundated an area of 140 km² and had a considerable human and economic impact. Three people lost their lives and 3500 had to evacuate their homes. Flood losses to residential, commercial and public assets were estimated to be 426 million Euro. Caldogno, a municipality with a population of about 11 000 located in the province of Vicenza, was among the most affected, with reported losses to those sectors reaching 25.7 million Euro (Regione del Veneto, 2011b). In this study, we adopt it as the second application case (Figure 2).

The data used for this application case are listed in Table II, and the results of individual model applications in terms of error statistics are shown in Table IV. The inundation characteristics were estimated using a coupled 1D/2D model of the study

area between the municipalities of Caldogno and Vicenza, and validated using data from sources such as aerial surveys and interviews with the local population. Building areas were derived from the cadastral map issued by the Veneto region. Building properties (i.e. building type, structural type, quality, number of floors and year of construction) were assessed through direct surveys to each damaged building. Building values were estimated based on data from the Chamber of Commerce of Vicenza. Losses to residential buildings were provided by the municipality of Caldogno and amount to a total of 7.55 million Euro. These correspond to actual restoration costs that were collected and verified within the scope of the loss compensation process by the State. Further details can be found in Scorzini and Frank (2015).

## 3. Ensemble construction and evaluation

Ensembles are finite sets of deterministic realisations of a random variable, whereby the prediction given by each ensemble member is assumed to represent an independent sample from an underlying true probability distribution (Hamill and Colucci, 1997). Ensembles can be used to account for various sources of uncertainty in physical processes, namely initial conditions, parameter and model uncertainty. The latter can be achieved by combining the output of different models to create a so-called multi-model ensemble (Weigel, 2012). In this section, we investigate how best to translate this concept to the field of flood loss modelling, and to which extent multi-model ensembles can improve the skill and usefulness of flood loss estimations.

### 3.1. Model rating

### 3.1.1. Method

The first challenge in constructing a multi-model ensemble to estimate flood loss for a certain future application is identifying models that are better suited to be participating members. One of the requirements for the construction of successful multi-model ensembles is that participating models are skilful; if a model is consistently worse than the others in terms of prediction quality, it should not be included (Hagedorn et al., 2005). Unfortunately, testing the level of skill of a model in predicting loss, for a certain type of asset and application setting, is often not possible. Such exercise would involve applying each candidate model to estimate loss for a past flood event with similar characteristics, and quantifying its performance based on past loss observations for the same assets. However, data required to perform such assessments are usually not available, as scarcity of data is still a major problem in the field of flood risk (Merz et al., 2010). Moreover, exposure and vulnerability tend to change over time, which is likely to affect loss estimates (Tanoue et al., 2016). Another issue of a more practical nature is that collecting, implementing and comparing flood loss models is laborious and time consuming. Because of the economic constraints that inevitably exist in any practical application, most users will likely have limited time to invest in that task. This becomes more problematic as the already large number of models available in the literature continues to increase.

A more practicable approach is to evaluate the suitability and potential performance of each model in estimating loss, for a given application setting, based on its properties. This is advantageous, as it does not require that each model be tested explicitly, and can instead be achieved by making use the information contained in a model metadata catalogue such as the ones developed by Gerl et al. (2016) or Pregnolato et al. (2015). However, models differ at various levels, and a model that is potentially superior regarding some of its properties may be inferior in terms of others (see Section 1). Consequently, directly evaluating the potential performance of flood loss models is arduous, and currently no established procedure exists to this end. In this subsection, we address this gap by proposing a framework to rate a set of flood loss models based on their properties. The framework is described as follows:

1. A probability tree of model properties is set up through expert elicitation. A set of $N$ independent properties that characterize flood loss models and that are likely to be informative for model performance are identified (e.g., damage metric). For each property (i.e. tree node) $n$, a set of mutually exclusive and collectively exhaustive categories are defined (e.g., relative and absolute). A subjective probability is then assigned to each category, corresponding to the degree of belief that a model that falls into that category will offer higher predictive performance than if it did in others. It follows that for each property $n$, the probabilities of the different categories sum to 1. Each path of the tree will have an associated probability that is obtained through the product of each node's probabilities $p_n$ along the path, therefore reflecting the degree of belief that that combination of model properties is the one that should be used;

2. Once the probability tree is set up, it can be used to assign scores to and rank flood loss models. Because the tree covers the entire space of possible categories within each property, all flood loss models will necessarily have a set of properties that matches one of the tree paths. Any model can thus be assigned a score that is equal to the probability of its respective path. When assigned to a certain number of *models* rather than to all the possible combinations of *model properties*, such scores no longer have a specific probabilistic meaning, nor are they intended to. Instead, the scores of different candidate models in a pool can be used to establish a relative degree of belief among them. This effectively provides users with information on their potential performance, in relation to the other models in the pool, through a structured and simple to use procedure.

### 3.1.2. Application

We apply this framework to the models and test cases presented in Section 2. We first propose a probability tree referring to flood loss models for buildings. It condenses expert knowledge and current state of the art in flood vulnerability of buildings, as well as experience from previous model transfer studies. The selection of properties and categories aims to balance comprehensiveness, objectivity and simplicity. Figure 3 presents the different properties, a succinct justification of their potential relevance in assessing model performance, and the respective categories and assigned subjective probabilities. Note that the maximum partial score that can be assigned to a model for properties 1 and 2 (shown in Figure 3) depends not only on the model but also on the hazard and exposure data sets. For example, if in a certain application case only water depth

data is available, loss models that use additional explanatory variables (e.g., velocity) should not be rated higher. We then use this setup to rate the flood loss models. The results are shown in Tables V and VI.

While model properties are expected to be informative for performance, they are not presumed to explain it fully. However, if model properties do have usefulness in assessing the performance of models in relation to one other, some degree of correlation between model scores and different performance metrics should exist. We evaluate this using the Spearman's rank correlation coefficient $r_s$ respectively between the scores shown in Tables V and VI and the error metrics shown in Tables III and IV. Results show a significant strong negative correlation between the variables ($-0.79 < r_s < -0.51$, $p < 0.01$), which suggests that model rating based on expert judgement is indeed informative for model performance. Note that no attempt was made to maximize correlations by fine-tuning the subjective probabilities, as not only would those not correspond to the experts' degrees of belief, but more importantly, because that would be no more than an exercise in overfitting to these two case studies. This topic is revisited in Section 3.2.2.

## 3.2. Ensemble-mean performance

The objective of the analyses presented in this section is twofold: to assess to which extent ensemble-means are able to improve skill in the estimation of flood losses, and to investigate how such ensembles should be constructed. Regarding the latter, two questions require particular attention: firstly, which and how many models to include as participating members, and secondly, how to weight those members. Both the ensemble size and the model weighting scheme are likely to have an effect on skill.

### 3.2.1. Based on model rating

In this exercise, the models and application cases described in Section 2 are used. For the construction of the various multi-model ensembles, we mimic the most common practical situation whereby it is necessary to estimate losses for a certain scenario for which past observational data is not available. Because in such situation, the skill of the individual models is not known, the potential suitability of each model for inclusion in a multi-model ensemble is evaluated through their properties, following the framework proposed in Section 3.1. Accordingly, different ensembles with increasing number of members are built, by including models sequentially from highest to lowest scores, according to Tables V and VI. Models with the same score are added to the ensemble simultaneously. The ensembles of different sizes constructed for each case study are shown in the x-axes of Figure 4, where 1 refers to the highest-ranked single model.

Losses given by ensemble means are estimated using two approaches: firstly, by assigning equal weights to all models, and secondly, by weighting them differently. Concerning this point, we now present some considerations. In the construction of an equal-weighted multi-model ensemble, the underlying hypothesis is that each model is independent and equally skilful, whereas this condition is most often not satisfied. For this reason, adopting different weights may increase the quality of multi-model predictions. However, finding optimal weights is not straightforward, and previous studies show that weighting models differently may result in different outcomes ranging from slight increases to degradation in performance (Doblas-

Reyes et al., 2005; Hagedorn et al., 2005; Knutti et al., 2010). Here, we aim to assess how weights affect ensemble-mean performances in estimating flood loss, again by reproducing a practical situation where the skill of models in a certain future application is not known. Therefore, assigned weights instead reflect the user's confidence in each model (Marzocchi et al., 2015). Because the framework proposed in Section 3.1 provides scores that are proportional to relative degrees of belief among models, in principle they may be used as weights. This is achieved by normalizing the weights of the participating models in each ensemble so that they sum to 1 (Spillatura, 2014). As mentioned in Section 2.1, in this study we aimed to ensure model independence by selecting a set of models developed independently, by different authors, using non-overlapping datasets (Cotton et al., 2006; Palmer et al., 2004). We therefore assume that possible model dependences are not relevant and have no bearing on the weighting scheme. Section 3.2.2 further discusses the effect of model weighting on ensemble-based loss estimation.

Ensemble-mean performances are calculated in terms of RMSE and MBE, which are shown in Figure 4 for the ensembles of different sizes – starting with a single model, the highest ranked for each case – and using the two weighting schemes described above. A number of observations can be made from this figure. Firstly, multi-model ensembles of any size, built by adding models with the highest degrees of belief first, considerably outperform the highest ranked single model in terms of both RMSE and MBE. This is observed for both application cases, the only exception being the MBE of some ensembles in the Caldogno case. Secondly, the performances obtained using the two different weighting approaches is mixed; while in some cases there is improvement by weighting ensemble members differently, in others the opposite is observed. The weighting approach generally does not have a significant impact on error metrics, especially when compared to the model selection. Thirdly, in both cases, the largest improvements in ensemble-mean performances are obtained after the first few highest ranked models are added. In relative terms, the impact of including additional models after that is lower. For example, in the Mulde and Caldogno case studies, the best performances are obtained with ensembles using respectively the highest-scoring four and six models. From a practical point of view, this is a particularly interesting finding because, as mentioned previously, it may not be feasible to implement a large number of models, and users may therefore be interested in parsimonious ensembles with the least number of models that lead to high predictive skill. However, in terms of probabilistic estimates of loss, smaller ensembles are less useful, which also needs to be taken into account when deciding on which ensemble size to use, as further discussed in Section 3.3.

Note that from here on, the equal-weighted expert-based multi-model ensembles shown in Figure 4 will be used as a basis for other analyses and further discussion, and for the sake of brevity will be referred to as EEM-ensembles.

Some of the above observations draw comparisons between multi-model ensembles and individual models, for which the highest ranked single model is used as reference. Even though that model may not necessarily correspond to the highest performing model (which it does not in either of the application cases used here; see Tables III/IV and V/VI), in a practical application case, users have no way of knowing which model is the 'best'. The above results very clearly demonstrate that in such situation, using a multi-model ensemble is preferable. However, it is also insightful to assess how the constructed multi-model ensembles perform in relation to the other single models. Therefore, in Figure 5, the error metrics of the predictions

given by EEM-ensemble-means and single models are presented, showing that the former consistently outperform the latter. Note that ensembles are not expected to outperform every single model in every possible situation, and it is possible that in some application cases, certain models have such high accuracy that combining them with other models results in lower performances. The problem is that it is usually not possible to identify such models beforehand. For example, in the Mulde

case, the Luino model slightly outperforms the constructed ensembles in terms of RMSE. This model consists in a simple stage-damage function that refers to a single building type, and was derived from data relative to a flood in Italy. Therefore, it is not expectable that it would consistently perform as well if applied to other analogous case studies. Overall, better performances should be obtained by using multi-model ensembles (Hagedorn et al., 2005).

### 3.2.2.   Based on simulated non-informativeness

The framework proposed in Section 3.1 and the subjective probabilities proposed in Figure 3 provide a basis for model selection and weighting in the development of multi-model ensembles. In Section3.2.1, we constructed various ensembles using this approach and evaluated their performance in estimating loss. However, in principle, it is possible that multi-model ensembles developed differently, i.e. by selecting different models and/or assigning different weights, would have higher skill. To investigate this issue, we simulate a so-called state of non-informativeness in terms of model suitability. This

consists in assuming we have no knowledge about how particular model characteristics might affect model predictive performance (Scherbaum and Kuehn, 2011), and therefore have no way of rating models. Accordingly, we implement a probabilistic sampling procedure that, for a large number of realisations, randomly generates weights for individual models regardless of their properties. On this basis, model ensembles are built and their predictive performance is calculated for the Mulde and the Caldogno case studies. The weight generation follows the stick-breaking method, whereby models are first

randomly ordered and then assigned weights sequentially. For each model, the weight is drawn from a continuous uniform distribution with a minimum value of 0 and a maximum value of 1 minus the sum of weights that have already been assigned. This approach, based on a large number of realisations, aims to cover all possible ensembles that can be constructed using the twenty flood loss models from Table I, using not only different weighting approaches (i.e. ensemble members weighted both equally and differently) but also different combinations of models. The latter is because according to

the stick-breaking method, once the model weights sum to 1, all other models receive a weight of 0 and are thus not included in the ensemble.

Scatter plots of the RMSE and MBE that result from the above procedure are presented in Figure 6 for both case studies. The same error metrics regarding the EEM-ensembles and the single models are also included. The plots show that a wide range of possible outcomes in terms of RMSE and MBE exist when random weights are assigned to models within the framework

of a state of non-informativeness. While the lower bounds of the resulting convex hull are defined by the error metrics of the lowest-performing models, the upper bounds (i.e. highest performances) are given not by any single model, but instead by multi-model ensembles, as expected. In this regard, it is clear that the model rating framework based on expert judgement and subjective probabilities proposed in Section 3.1 add value to the ensemble development process. Indeed, ensembles that

are constructed by adding models prioritized in terms of potential suitability (shown in Figure 4) are among the highest performing ensembles, considering all the existing possibilities. It is interesting to highlight that the simple unweighted mean of all models also performs relatively well, which suggests that if no knowledge is available on model properties and/or on how they influence performance, it is better to include all models than to wrongly select them.

The plots also show that it is possible to create certain ensembles that lead to better skill in relation to the ones developed based on expert judgement. However, the potential relative degree of improvement is very low in both test cases, more markedly so in the Caldogno case, which reinforces the idea that the approach proposed in Section 3.1 provides a good basis for ensemble construction. We do not attempt to maximize the performance of the constructed multi-model ensembles based on the results obtained in this exercise, as this would be of little relevance. Analogously to the 'best' model discussion in

Section 3.2.1, in a practical application the ensembles cannot be tested beforehand. Finding specific weights that maximize performance for the Mulde and the Caldogno case studies would consist in pointless overfitting, as such weights necessarily vary from case to case. In addition, it is likely that such weights would not make sense from the perspective of an expert. Instead, the objective here is that ensembles are constructed in a manner that leads to good performances in all situations, which the results support. Finally, Figure 6 corroborates that correctly selecting models for an ensemble is more important

than weighting them. The EEM-ensembles, which result from model selection only, display error metrics close to the minimum obtainable from a wide range of possible outcomes. In comparison, further improvements that could possibly be achieved by assigning different weights to ensemble members are very small.

## 3.3.    Probabilistic application

In Section 3.2, multi-model ensemble-means have been shown to provide more skilful estimates of flood losses than single

models. Another motivation for the use of such ensembles is that they may be used to quantify model uncertainty and obtain probabilistic distributions of possible outcomes rather than single point estimates, which is, as discussed previously, required for optimal decision-making. In this section, we offer some discussion on this topic.

It is first necessary to make clear what the probabilistic meaning of a multi-model ensemble is. Multi-model ensembles do not directly provide probability distributions of a certain variable; instead, ensemble predictions are *a priori* only finite sets

of deterministic realisations of that variable. The question then arises how a probability distribution can be obtained from such ensembles. The simplest approach is to adopt a frequentist interpretation of the ensembles, whereby the probability of a certain event to happen is estimated by the fraction of ensemble members predicting it. However, such approach can only produce reasonable probabilistic estimates if many ensemble members are available. Better probabilistic estimates may in principle be obtained by dressing the ensemble members with kernel functions or by fitting a suitable parametric distribution

to them, provided that this is done in an appropriate manner (Weigel, 2012).

### 3.3.1. Skill and reliability

Regardless of the method that is used to obtain probabilistic estimates from multi-model ensembles, it is first important to evaluate the 'raw' ensembles, with minimum interference from the ensemble interpretation model that is used. This can be achieved using the continuous ranked probability score (CRPS) (Bröcker, 2012; Hersbach, 2000). We calculate the CRPS for the EEM-ensembles, and present the results in Figure 7. This is done for the Caldogno case study, as the low number of data points in the Mulde case (19) are insufficient for such analysis.

The probabilistic skill of the ensembles is observed to have an increasing trend (i.e. decreasing CRPS) with the number of participating members. This is to some extent expected, as ensemble size is known to have an effect on probabilistic skill scores, which is explained by the fact that probabilistic estimates derived from ensembles become more unreliable as the size of the ensemble gets smaller (Weigel, 2012). This highlights the need of using a considerable number of models when the objective is to obtain reliable (i.e. statistically consistent) probabilistic estimates of flood loss. Another requirement to achieve this is that the ensemble itself is reliable, in the sense that ensemble members and observations are sampled from the same underlying probability distributions or, in other words, that they are statistically indistinguishable from each other (Leutbecher and Palmer, 2008). Even an ensemble of infinite size is unable to yield reliable probabilistic estimates if its member are not reliable (e.g., if they are heavily biased). For illustration, we assess reliability considering an ensemble comprising all twenty models implemented in this study using the rank histogram, which is shown in Figure 8. As expected, the ensemble is not perfectly reliable; however, the counts do tend to oscillate around $\frac{n}{m+1} = \frac{296}{21}$, which suggests a reasonable degree of reliability. In addition, the ensemble appears to be slightly overdispersive, due to an overpopulation of central ranks of the histogram.

### 3.3.2. Loss estimation

Finally, we illustrate the simplest approach to obtain a probabilistic distribution of flood losses using a multi-model ensemble. For each building, a value of loss is randomly generated using the reverse transform sampling method, whereby a number $u \sim [0, 1]$ is sampled from the standard uniform distribution, and the corresponding quantile is sampled from the empirical cumulative distribution function (ECDF) of losses given by ensemble members through linear interpolation. The losses for each building are then summed, and a total loss is obtained. This process is repeated a large number of times, yielding a loss distribution for the flood event. The results for the Caldogno application, based on 10 000 realisations, are shown in Figure 9 in the form of a histogram and ECDF of total loss.

Statistical post-processing techniques may be used to improve the reliability of probabilistic predictions. This is common practice in the field of numerical weather prediction, for example. However, in that case, relatively long time series of past observational data for a certain variable (e.g., temperature) at a certain location are usually available, and such data continue to be collected, which allows the predictive system to be calibrated and the forecasts verified. This is in contrast with the case of flood loss estimations, where loss models necessarily need to be transferred due to the rarity of the events and the

difficulty in obtaining data. In the particular case of probabilistic loss estimates based on ensembles, it is therefore necessary to investigate how best to improve their reliability for future applications by considering data from previous flood events often occurring in different contexts. In addition, as mentioned previously, the reliability of probabilistic estimates may also be improved by using a more sophisticated ensemble interpretation method (i.e., kernel dressing or parametric distribution fitting). However, the most appropriate approach to do this in the case of flood loss modelling also needs to be investigated. These topics are beyond the scope of this article.

## 4. Conclusions

Flood loss modelling is associated with considerable uncertainty that is often neglected. In fact, most currently available flood loss models are deterministic, providing only single point estimates of loss. Users interested in performing a risk assessment will typically select one such model from the large number available in the literature, based on their perception of which one is the most suitable for the application case at hand. However, this is generally done rather arbitrarily. Moreover, the uncertain nature of flood loss estimations means that the performance of any single deterministic model may vary considerably from case to case, as large disparities in model outcomes exist even among apparently comparable models. This approach is therefore flawed at two main levels: first, flood risk estimates are highly sensitive to the selection of the flood loss model, and second, deterministic estimates of loss do not lead to optimal decision-making. In this study, we have proposed a novel approach to tackle these issues and advance the state of the art in flood loss modelling, based on the application of the concept of multi-model ensembles. This technique, which is widely used in fields such as weather forecasting, consists in combining the outcomes of different models in order to improve prediction skill and sample model uncertainty.

In order to support ensemble construction, we have first proposed a framework to assess the suitability of flood loss models to specific application cases, based on some of their main properties, through expert knowledge. This approach is advantageous as it does not require that all candidate models are implemented beforehand, which is often not achievable in practice. Based on such framework, we have proposed a scoring scheme for flood loss models for residential buildings, and applied it to the twenty models and two applications cases used in this study. The obtained model scores show significant strong negative rank correlation with error metrics, suggesting that the proposed approach is useful, and that expert judgement is informative for model performance and selection.

The constructed ensembles have been shown to considerably outperform the highest ranked single models in the estimation of flood losses. This demonstrates that in a practical application, where model performances cannot be tested beforehand, using multi-model ensembles will result in more skilful loss estimates. Ensemble-means were also tested against all single models, consistently showing higher accuracy. Equal-weighted ensembles generally displayed performances comparable to the score-weighted ones. The largest improvements in ensemble-mean performances were observed after the first few highest ranked models were added to the ensembles, which is a useful finding for practical applications, where it is not always feasible to implement a large number of models. We have also simulated a state of non-informativeness and

randomly generated a large set of multi-model ensembles, representative of all possible ensembles that can be constructed using the twenty flood loss models adopted in this study. The ensembles based on expert-based scoring approach were among the most skilful, highlighting its value in the construction of multi-model ensembles. Results also suggest that model selection is more important than weighting. Further insight may be gained by testing the approach in other application cases

5  and using a different set of flood loss models.

Larger ensembles showed higher probabilistic skill than smaller ones, which results from the increased intrinsic unreliability of ensembles as the number of participating members decreases. Therefore, if on the one hand only a limited number of models is necessary to obtain accurate mean estimates of loss, on the other additional effort in model implementation is recommended when the objective is to derive a probabilistic distribution of loss that captures model uncertainty. For the

10  Caldogno case study, we have illustrated how such a distribution can be constructed, adopting a simple equal-weighted ensemble comprising all twenty models. The results demonstrate that the use of multi-model ensembles represents a simple and pragmatic way of obtaining reliable flood loss distributions, which are more useful for decision-making than single point estimates of loss. Reliability may be further improved by calibrating the ensembles and/or adopting more sophisticated ensemble interpretation models, which warrants further research.

**Acknowledgements**

This research was partly supported by the European Union's Horizon 2020 research and innovation programme, through the IMPREX project (Grant Agreement number 641811) and the H2020 Insurance project (Grant Agreement number 730459). Further support has been received from Guy Carpenter and Company Ltd. (www.guycarp.com).

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

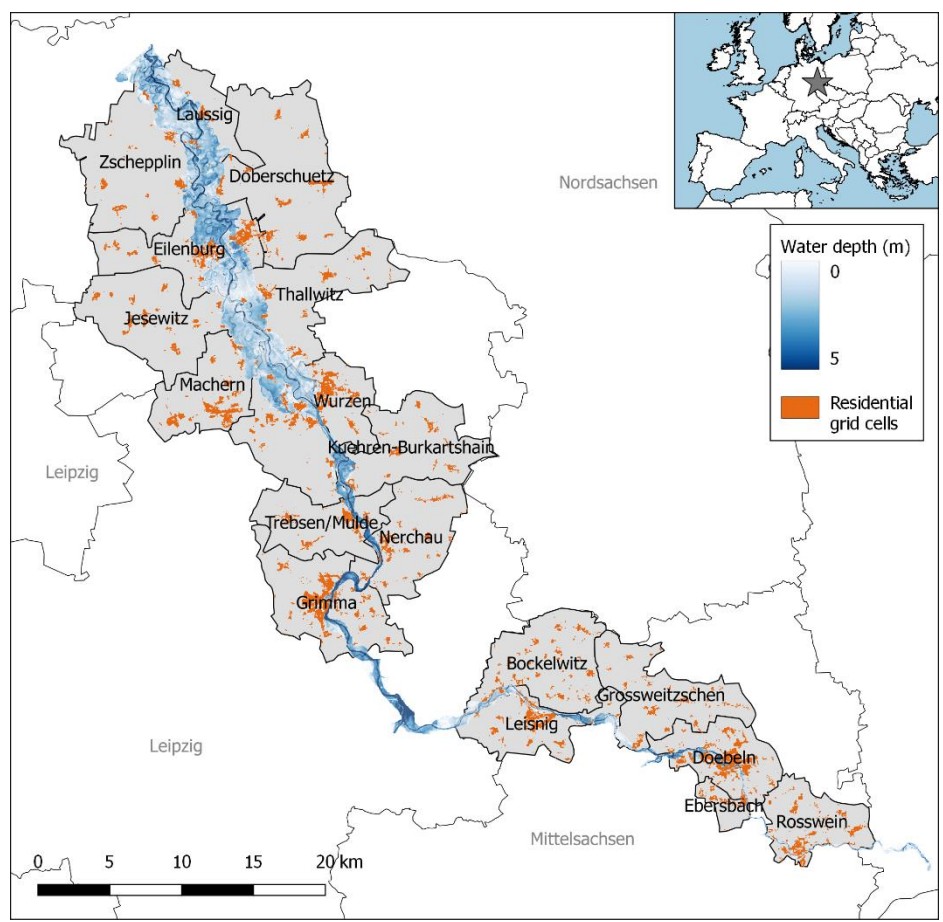

**Figure 1: 2002 flood along the Mulde River, in Germany. The figure shows the municipalities considered in the case study (grey), the estimated flood extension and water depths (blue), and the location of the residential grid cells (orange).**

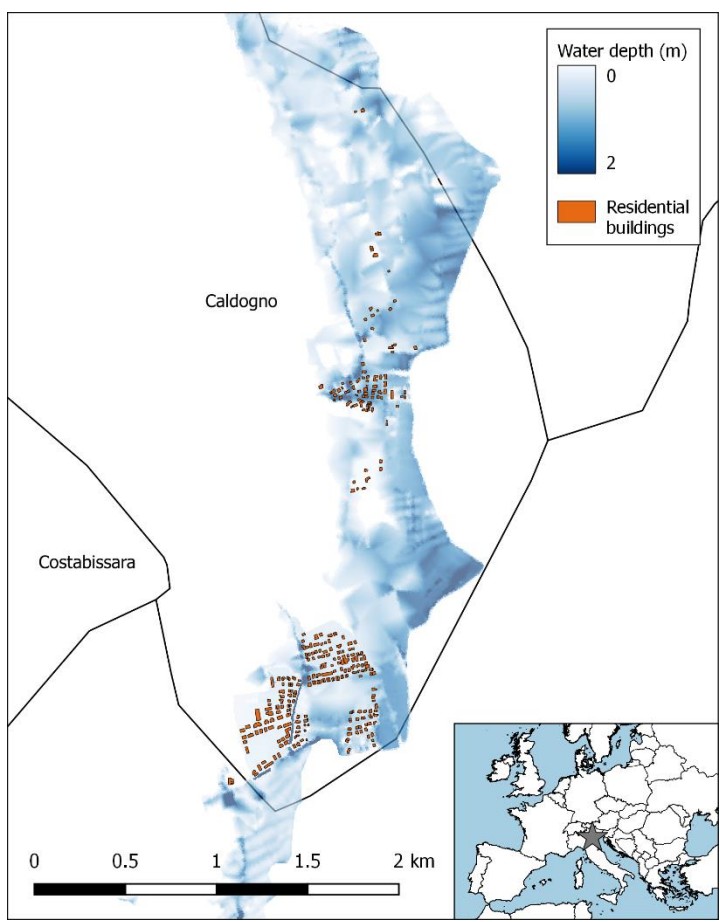

**Figure 2: 2010 Bacchiglione river flood in Caldogno, Italy. The figure shows the estimated flood extension and water depths (blue), and the location of the residential buildings considered in the study (orange).**

**1. Flood intensity measures**

*Flood damage processes are influenced by multiple factors. Although water depth is considered the most important intensity measure, additional variables tend to improve model predictive skill.*

| | |
|---|---|
| Water depth and additional variables | 0.65 |
| Water depth only | 0.35 |

**2. Characterization of exposed assets**

*The degree of characterization of assets in flood loss model is directly related with potential performance, as insufficient distinction may result in their inappropriate application to assets for which they are not suited.*

| | |
|---|---|
| Building type and physical properties (e.g. material, no. of floors) | 0.45 |
| Building type only (e.g. single family house) | 0.35 |
| Occupancy type only (e.g. residential) | 0.20 |

**3. Similarity of local context with application setting**

*Models generally perform better in regions with a socio-economic context comparable to the one for which they were developed, as they tend to have more similarities in terms of construction quality and practices.*

| | |
|---|---|
| Same region | 0.40 |
| Same country | 0.30 |
| Same WESP classification[1] | 0.20 |
| Different WESP classification[1] | 0.10 |

[1] According to UN World Economic Situation and Prospects 2016. A: developed economies; B: economies in transition; C: developing countries; D: least developed countries.

**4. Flood type in relation to application setting**

*Flood loss models are usually constructed for specific types of inundation (e.g., fluvial flood), and will usually perform worse when applied to flood events with different dominant damaging processes.*

| | |
|---|---|
| Identical | 0.70 |
| Different | 0.30 |

**5. Damage metric**

*Relative loss models, which express loss as a fraction of the total asset value, offer better transferability, whereas absolute loss models have little applicability outside the specific case for which they were developed.*

| | |
|---|---|
| Relative | 0.70 |
| Absolute | 0.30 |

**Figure 3: Proposed set of properties (probability tree nodes) that are considered relevant to assess the performance of flood loss models for buildings, and respective categories and subjective probabilities.**

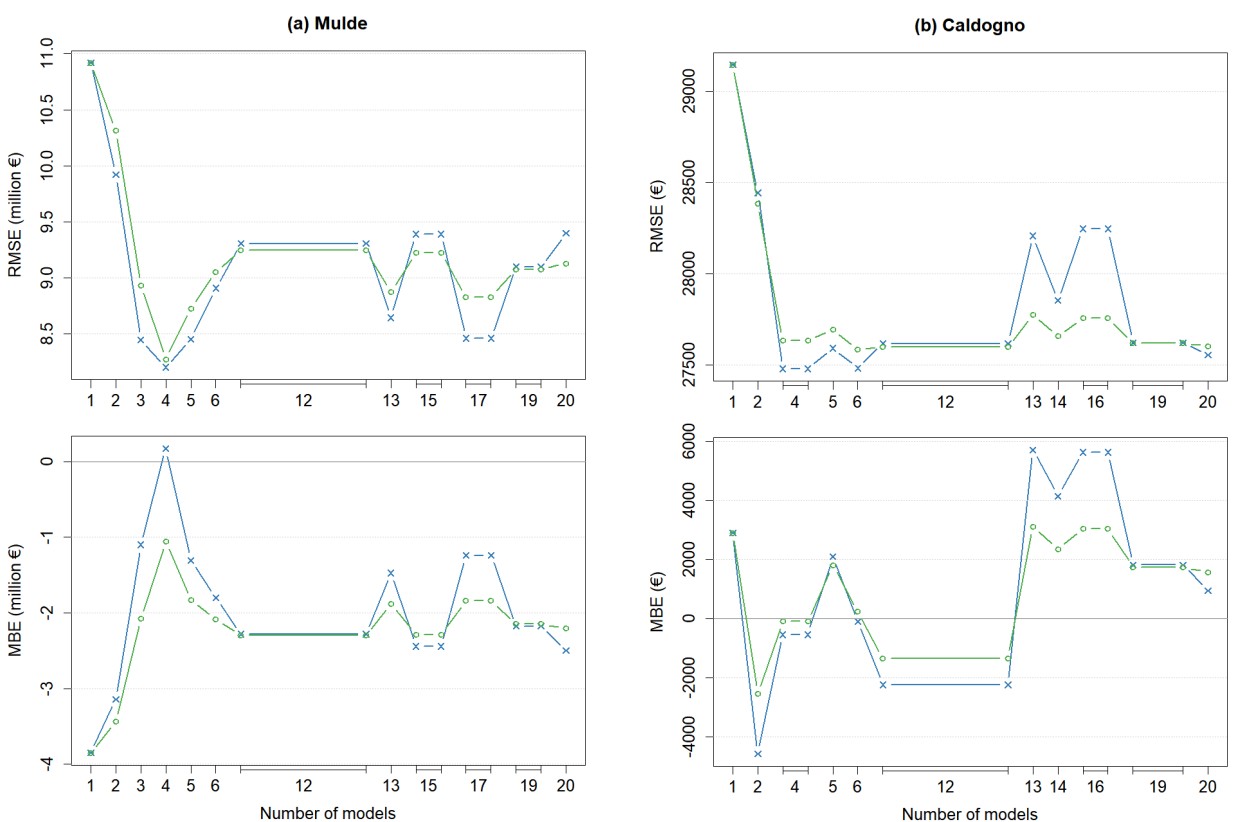

**Figure 4: Root mean square error (RMSE) and mean bias error (MBE) of the means of ensembles of increasing size, with models included sequentially from highest to lowest score, starting with the highest ranked single model. Blue crosses and green circles refer to ensembles weighted equally and differently, respectively.**

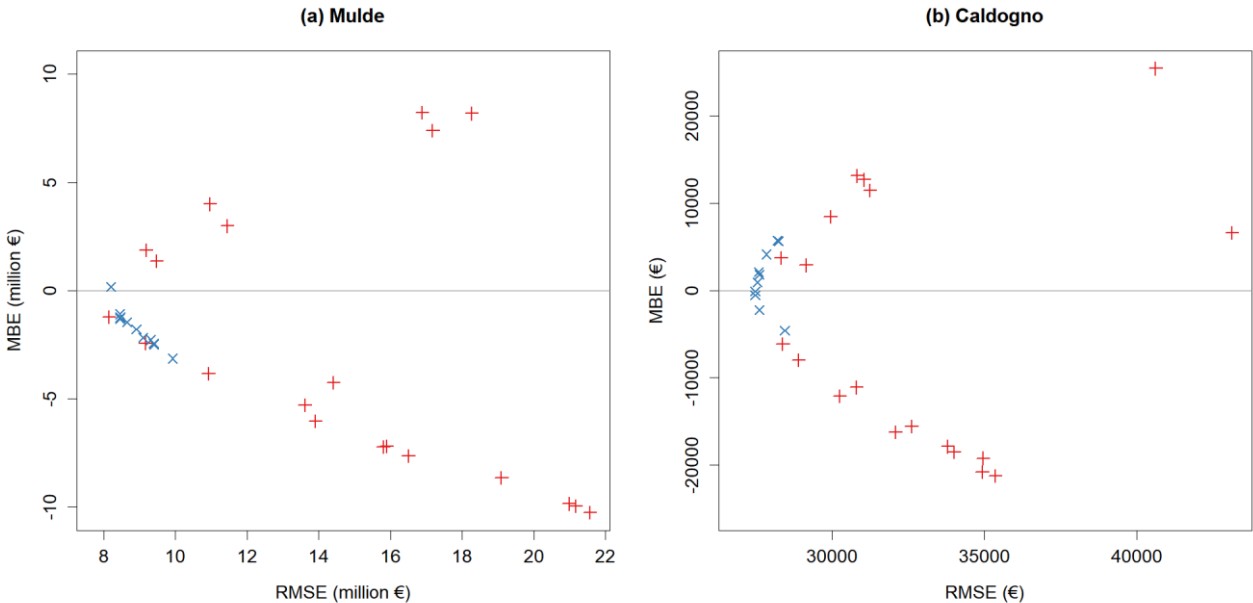

**Figure 5: Root mean square error (RMSE) and mean bias error (MBE) of the EEM-ensemble means, represented by blue crosses, and single models, by red plus signs.**

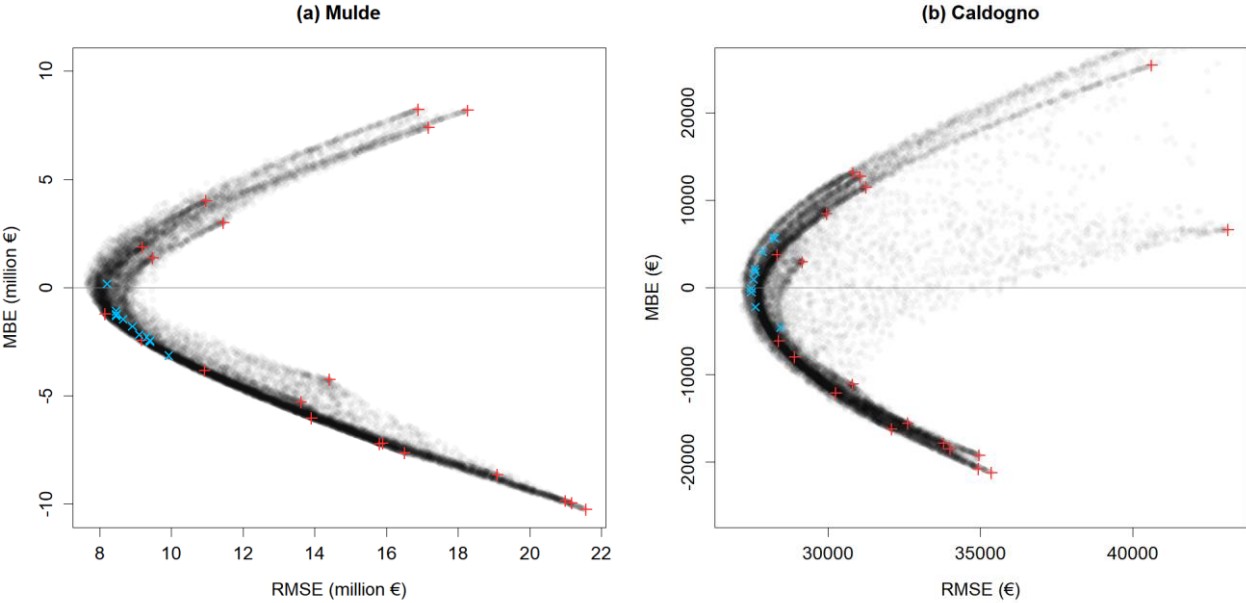

**Figure 6: Root mean square error (RMSE) and mean bias error (MBE) of 20 000 multi-model ensemble means, generated by simulating a state of non-informativeness, whereby each participating member is assigned a random weight. Blue crosses and red plus signs refer respectively to the EEM-ensembles and the single models.**

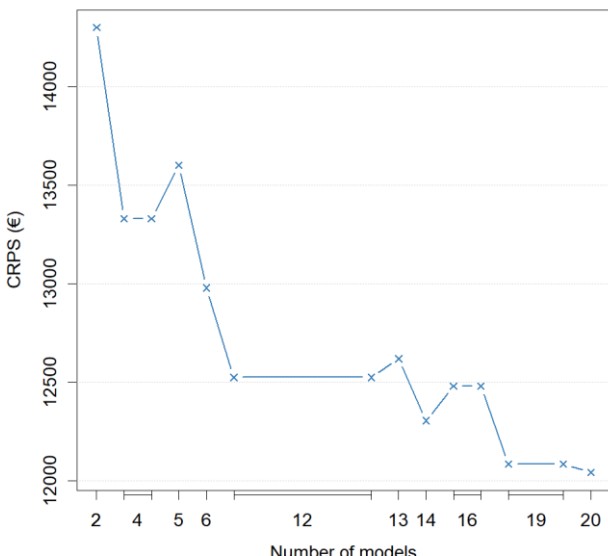

**Figure 7: Continuous ranked probability score (CRPS) of the EEM-ensembles for the Caldogno application case.**

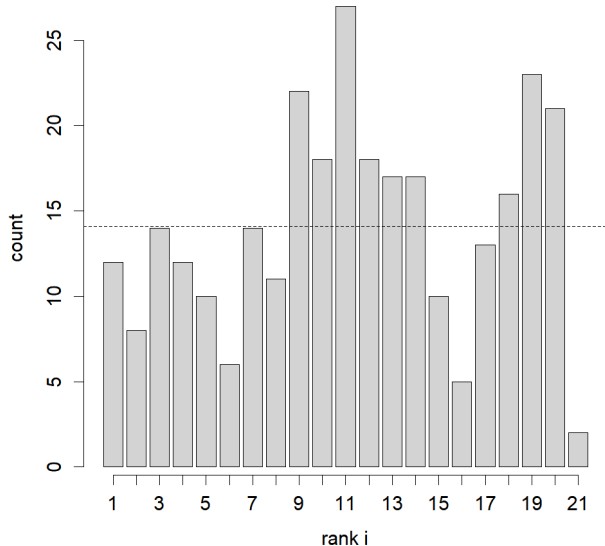

**Figure 8: Rank histogram relative to the 20-model ensemble for the Caldogno application case.**

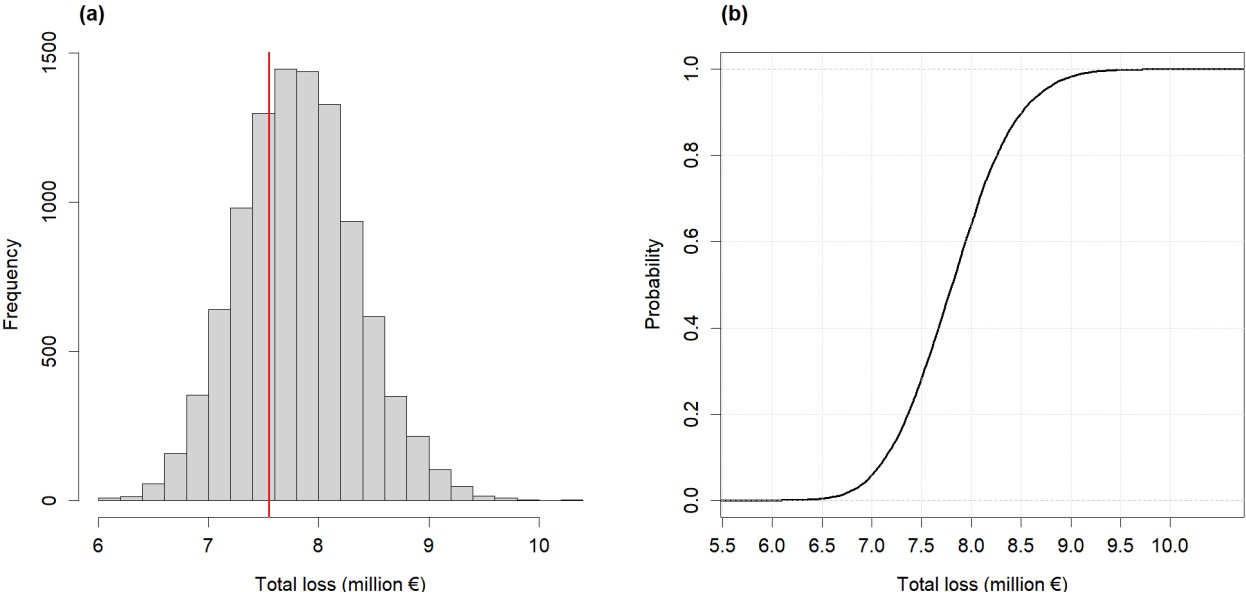

**Figure 9: Probabilistic estimates of total loss, relative to model uncertainty, for the Caldogno application case, based on 10 000 realisations of loss to each building. (a) Histogram, with observed loss shown by the vertical red line. (b) Empirical cumulative distribution function (ECDF).**

**Table I: Models included in this study, including some of their properties.**

| Name | Hazard variables[a] | Exposure variables[b] | Country | Region/Catchment | Flood type | Damage metric | Reference |
|---|---|---|---|---|---|---|---|
| ANUFlood | wd | fa | Australia | - | fluvial | absolute | Department of Natural Resources and Mines (2002) |
| Budiyono | wd | bt | Indonesia | Ciliwung River | fluvial | relative | Budiyono et al. (2015) |
| DSM | wd | bt | The Netherlands | - | fluvial, coastal | relative | Klijn et al. (2007) |
| Dutta | wd | str | Japan | Ichinomiya river basin, Chiba prefecture | fluvial | relative | Dutta et al. (2003) |
| FLEMO | wd, con, rp | bt, bq, pre | Germany | Elbe, Danube | fluvial | relative | Elmer et al. (2010) |
| HAZUS-MH | wd | bt, nf, bas | USA | - | fluvial, coastal | relative | Scawthorn et al. (2006) |
| HOWAS | wd | bt, bas | Germany | - | fluvial | absolute | Buck and Merkel (1999) |
| HWS-GIS | wd | - | Germany | Lippe | fluvial | relative | Hydrotec (2002) |
| ICPR | wd | - | Switzerland, Germany, France, Netherlands | Rhine | fluvial | relative | ICPR (2001) |
| IKSE | wd | - | Germany | Elbe | fluvial | relative | IKSE (2003) |
| Luino | wd | - | Italy | Boesio basin, in the Lombardy Region | fluvial | relative | Luino et al. (2009) |
| MCM | wd, id | bt | England, Wales | - | fluvial, coastal | absolute | Penning-Rowsell et al. (2005) |
| MERK | wd | nf, bas | Germany | Coast of Schleswig-Holstein | coastal | relative | Reese et al. (2003) |
| Pistrika&Jonkman | wd, fv | - | USA | Mississippi River | fluvial, levee breach | relative | Pistrika and Jonkman (2010) |
| Riha&Marcikova | wd, id | bt, oth | Czech Republic | - | fluvial | relative | Riha and Marcikova (2009) |
| Toth | wd | bt, str, nf | Hungary | Körös corner flood area | fluvial | relative | Tóth et al. (2008) |
| TYROL | wd | - | Austria | Tyrol | fluvial | absolute | Huttenlau et al. (2010) |
| Vanneuville | wd | bt | Belgium | - | fluvial | relative | Vanneuville et al. (2006) |
| Vojinovic | wd | fa | St Maarten | - | fluvial | absolute | Vojinovic et al. (2008) |
| Yazdi&Neyshabouri | wd | - | Iran | Kan basin | fluvial | relative | Yazdi and S. Neyshabouri (2012) |

[a] Hazard variables: wd: water depth; fv: flow velocity; id: inundation duration; con: contamination; rp: return period.

[b] Exposure variables: bt: building type; str: building structure; bq: building quality; nf: number of floors; bas: presence of basement; fa: floor area; pre: precautionary measures.

**Table II: Input variables for the Mulde and Caldogno application cases.**

| Component | Variable | Resolution | |
|---|---|---|---|
| | | **Mulde** | **Caldogno** |
| Hazard | Water depth (m) | $10\times10$ m$^2$ grid cell | 5x5 m$^2$ grid cell |
| | Flow velocity ([a]) | Municipality | 5x5 m$^2$ grid cell |
| | Inundation duration (h) | Municipality | - |
| | Return period (yr) | Catchment | - |
| | Contamination indicator | Municipality | - |
| Exposure | Building floor area (m$^2$) | Municipality [b] | Building |
| | Value (€) | $10\times10$ m$^2$ grid cell | Building |
| | Building type | Municipality | Building |
| | Building quality | Municipality | Building |
| | Building structure | - | Building |
| | Number of floors | - | Building |
| | Presence of basement | - | Building |
| | Year of construction | - | Building |
| | Precautionary measures indicator | Municipality | - |
| Loss | Reported loss (€) | Municipality | Building |

[a] indicator for Mulde; m.s$^{-1}$ for Caldogno. [b] Mean value.

**Table III: Results of individual model applications in the Mulde case: root mean square error (RMSE) and mean bias error (MBE), sorted by RMSE.**

| Model name | Error metrics (million €) | |
| | RMSE | MBE |
| --- | --- | --- |
| Luino | 8.143 | -1.230 |
| IKSE | 9.160 | -2.433 |
| Dutta | 9.177 | 1.870 |
| DSM | 9.469 | 1.359 |
| FLEMO | 10.918 | -3.850 |
| HAZUS-MH | 10.964 | 3.998 |
| Riha&Marcikova | 11.449 | 2.986 |
| Vanneuville | 13.608 | -5.302 |
| Toth | 13.906 | -6.050 |
| MCM | 14.405 | -4.266 |
| HWS-GIS | 15.796 | -7.237 |
| ICPR | 15.888 | -7.201 |
| MERK | 16.497 | -7.656 |
| Pistrika&Jonkman | 16.883 | 8.235 |
| Yazdi&Neyshabouri | 17.174 | 7.398 |
| Budiyono | 18.258 | 8.190 |
| Vojinovic | 19.095 | -8.667 |
| HOWAS | 20.982 | -9.863 |
| TYROL | 21.160 | -9.979 |
| ANUFlood | 21.559 | -10.273 |

**Table IV: Results of individual model applications in the Caldogno case: root mean square error (RMSE) and mean bias error (MBE), sorted by RMSE.**

| Model name | Error metrics (€) | |
| --- | --- | --- |
| | RMSE | MBE |
| IKSE | 28324.2 | 3742.0 |
| Toth | 28381.9 | -6154.0 |
| HWS-GIS | 28901.2 | -7974.5 |
| FLEMO | 29147.5 | 2899.9 |
| DSM | 29950.1 | 8437.0 |
| Riha&Marcikova | 30248.8 | -12084.7 |
| MCM | 30798.3 | -11106.9 |
| HAZUS-MH | 30829.7 | 13131.2 |
| Luino | 31050.4 | 12688.3 |
| Dutta | 31242.9 | 11470.0 |
| MERK | 32078.9 | -16228.1 |
| Vojinovic | 32605.8 | -15581.1 |
| TYROL | 33798.5 | -17867.9 |
| ANUFlood | 34010.5 | -18510.8 |
| Vanneuville | 34925.9 | -20809.0 |
| HOWAS | 34954.3 | -19213.4 |
| ICPR | 35356.4 | -21224.3 |
| Yazdi&Neyshabouri | 40614.3 | 25441.6 |
| Budiyono | 43112.8 | 6602.7 |
| Pistrika&Jonkman | 109444.7 | 101296.1 |

**Table V: Model scores for the Mulde application case.**

| Model name | Node probabilities | | | | | Score ($10^{-2}$) | Score rank |
|---|---|---|---|---|---|---|---|
| | $p_1$ | $p_2$ | $p_3$ | $p_4$ | $p_5$ | | |
| FLEMO | 0.65 | 0.35 | 0.30 | 0.70 | 0.70 | 3.34 | 1 |
| IKSE | 0.35 | 0.20 | 0.40 | 0.70 | 0.70 | 1.37 | 2 |
| Riha&Marcikova | 0.65 | 0.20 | 0.20 | 0.70 | 0.70 | 1.27 | 3 |
| HAZUS-MH | 0.35 | 0.35 | 0.20 | 0.70 | 0.70 | 1.20 | 4 |
| HWS-GIS | 0.35 | 0.20 | 0.30 | 0.70 | 0.70 | 1.03 | 5 |
| MCM | 0.65 | 0.35 | 0.20 | 0.70 | 0.30 | 0.96 | 6 |
| DSM | 0.35 | 0.20 | 0.20 | 0.70 | 0.70 | 0.69 | 7 |
| Dutta | 0.35 | 0.20 | 0.20 | 0.70 | 0.70 | 0.69 | 7 |
| ICPR | 0.35 | 0.20 | 0.20 | 0.70 | 0.70 | 0.69 | 7 |
| Luino | 0.35 | 0.20 | 0.20 | 0.70 | 0.70 | 0.69 | 7 |
| Toth | 0.35 | 0.20 | 0.20 | 0.70 | 0.70 | 0.69 | 7 |
| Vanneuville | 0.35 | 0.20 | 0.20 | 0.70 | 0.70 | 0.69 | 7 |
| Pistrika&Jonkman | 0.65 | 0.20 | 0.20 | 0.30 | 0.70 | 0.55 | 13 |
| HOWAS | 0.35 | 0.20 | 0.30 | 0.70 | 0.30 | 0.44 | 14 |
| MERK | 0.35 | 0.20 | 0.30 | 0.30 | 0.70 | 0.44 | 14 |
| Budiyono | 0.35 | 0.20 | 0.10 | 0.70 | 0.70 | 0.34 | 16 |
| Yazdi&Neyshabouri | 0.35 | 0.20 | 0.10 | 0.70 | 0.70 | 0.34 | 16 |
| ANUFlood | 0.35 | 0.20 | 0.20 | 0.70 | 0.30 | 0.29 | 18 |
| TYROL | 0.35 | 0.20 | 0.20 | 0.70 | 0.30 | 0.29 | 18 |
| Vojinovic | 0.35 | 0.20 | 0.10 | 0.70 | 0.30 | 0.15 | 20 |

**Table VI: Model scores for the Caldogno application case.**

| Model name | Node probabilities | | | | | Score ($10^{-2}$) | Score rank |
|---|---|---|---|---|---|---|---|
| | $p_1$ | $p_2$ | $p_3$ | $p_4$ | $p_5$ | | |
| FLEMO | 0.65 | 0.35 | 0.20 | 0.70 | 0.70 | 2.23 | 1 |
| Riha&Marcikova | 0.65 | 0.20 | 0.20 | 0.70 | 0.70 | 1.27 | 2 |
| HAZUS-MH | 0.35 | 0.35 | 0.20 | 0.70 | 0.70 | 1.20 | 3 |
| Toth | 0.35 | 0.35 | 0.20 | 0.70 | 0.70 | 1.20 | 3 |
| Luino | 0.35 | 0.20 | 0.30 | 0.70 | 0.70 | 1.03 | 5 |
| MCM | 0.65 | 0.35 | 0.20 | 0.70 | 0.30 | 0.96 | 6 |
| DSM | 0.35 | 0.20 | 0.20 | 0.70 | 0.70 | 0.69 | 7 |
| Dutta | 0.35 | 0.20 | 0.20 | 0.70 | 0.70 | 0.69 | 7 |
| HWS-GIS | 0.35 | 0.20 | 0.20 | 0.70 | 0.70 | 0.69 | 7 |
| ICPR | 0.35 | 0.20 | 0.20 | 0.70 | 0.70 | 0.69 | 7 |
| IKSE | 0.35 | 0.20 | 0.20 | 0.70 | 0.70 | 0.69 | 7 |
| Vanneuville | 0.35 | 0.20 | 0.20 | 0.70 | 0.70 | 0.69 | 7 |
| Pistrika&Jonkman | 0.65 | 0.20 | 0.20 | 0.30 | 0.70 | 0.55 | 13 |
| MERK | 0.35 | 0.35 | 0.20 | 0.30 | 0.70 | 0.51 | 14 |
| Budiyono | 0.35 | 0.20 | 0.10 | 0.70 | 0.70 | 0.34 | 15 |
| Yazdi&Neyshabouri | 0.35 | 0.20 | 0.10 | 0.70 | 0.70 | 0.34 | 15 |
| ANUFlood | 0.35 | 0.20 | 0.20 | 0.70 | 0.30 | 0.29 | 17 |
| HOWAS | 0.35 | 0.20 | 0.20 | 0.70 | 0.30 | 0.29 | 17 |
| TYROL | 0.35 | 0.20 | 0.20 | 0.70 | 0.30 | 0.29 | 17 |
| Vojinovic | 0.35 | 0.20 | 0.10 | 0.70 | 0.30 | 0.15 | 20 |