# Peer review of "Multi-model ensembles for assessment of flood losses and associated uncertainty"

_Natural Hazards and Earth System Sciences, 2017_

## Referee Comment (RC1) · Anonymous Referee #1 · 23 Nov 2017

The paper addresses an interesting research topic, evaluating the utility of using ensembles of multiple flood damage models to improve loss estimations and quantify the related uncertainties. The work is well structured and generally well presented. The analyses carried out provide good evidence of the benefits of using multi-model ensembles compared to the application of single damage models. I believe that the manuscript is worth of publication in NHESS, provided that Authors address a few issues.

Main points

- although the paper reads well, I believe that a better structure could improve its use-

fulness for the reader. In particular, most subsections of Section 3 include first a description of the work carried out, and then the results (e.g. Section 3.1 begins with the description of the methods used to build and evaluate the ensemble of models, followed by presentation and discussion of results) . I would suggest to separate the method descriptions from the presentation and discussion of results, putting them in different sections; this would improve the readibility of the paper and make easier the consultation.

- How did the Authors select the models for their work? The paper by Gerl et al (2016) reviews a larger number of models, so the Authors need to explain the criteria applied for their selection.

- An important point that I miss regards models availability. As a matter of fact, several flood damage models are hardly usable in practice, either because not accessible (e.g. commercial models), or because the publicly available information is incomplete and does not allow application (e.g. some research models). Are the models selected by the Authors freely accessible? This would be a major point to foster the use of multi-model ensembles as recommended by the Authors.

Minor points

- title of Section 3.3 is not much informative for the reader, please change it.

- Please indicate the measure unit in Tables 3 and 5.

- Table should be numbered according to the order of citation in the text.

- Descriptions at page 11, lines 12-30 are not completely clear to me (e.g. I did not understand what measure is used to build the rank histogram), could you please add some more details?

---

## Referee Comment (RC2) · Anonymous Referee #2 · 20 Feb 2018

**Improving accuracy and quantifying uncertainty in flood loss estimations through the use of multi-model ensembles**

[revised manuscript text omitted]
 have been adopted in this study. They were selected from the model metadata catalogue developed by Gerl et al. (2016), which fully describes each model in terms of their properties.

30    This information is necessary to assess model suitability according to the framework proposed in Section 3.1. For each model, the properties relevant for this study as well as the corresponding reference are shown in Table I. Each model is implemented to compute flood losses for the two application cases described in Section 2.2. This consists in the largest to

date application of different flood loss models within the scope of a scientific study on flood risk. In the estimation of losses for each asset, the best-matching function from each model is adopted. Losses given in absolute terms are adjusted for inflation. The modelled losses are provided as supplementary material, and performance metrics for each case are shown in Tables III and V. These are used in Section 3 in the evaluation of multi-model ensembles.

[revised manuscript text omitted]

**Table II: Input variables for the Mulde application case.**

[revised manuscript text omitted]

---

## Author Comment (AC1) · 8 Mar 2018

**Response to Referee 1**

We would like to thank the referee for the time and effort put into reviewing the manuscript. This response carefully addresses all the comments. Where applicable, changes are proposed to the manuscript accordingly. Following the guidelines of the NHESS Editorial Board, the revised manuscript was not prepared at this point.

*The paper addresses an interesting research topic, evaluating the utility of using ensembles of multiple flood damage models to improve loss estimations and quantify the related uncertainties. The work is well structured and generally well presented. The analyses carried out provide good evidence of the benefits of using multi-model ensembles compared to the application of single damage models. I believe that the manuscript is worth of publication in NHESS, provided that Authors address a few issues.*

*Main points*

*- although the paper reads well, I believe that a better structure could improve its usefulness for the reader. In particular, most subsections of Section 3 include first a description of the work carried out, and then the results (e.g. Section 3.1 begins with the description of the methods used to build and evaluate the ensemble of models, followed by presentation and discussion of results). I would suggest to separate the method descriptions from the presentation and discussion of results, putting them in different sections; this would improve the readability of the paper and make easier the consultation.*

We agree with the Reviewer that the structure of the manuscript can be improved. After careful consideration, we propose the following changes:

- Section 2 will be renamed to "Setup of validation exercise". A new subsection called "Evaluation methods" will be added. This will briefly describe and provide references on the evaluation methods adopted in the study, specifically: RMSE, MBE, CRPS and the Rank Histogram. These changes will aid overall clarity, as well as improve readability of Section 3.4.

- We believe that apart from the evaluation methods, the remainder of Section 3 follows a line of thought where the justification for each step of our study is made clearer after the previous step has been presented, applied, and discussed. In order to facilitate consultation and improve readability, we instead propose to restructure this section as follows:

3.      Ensemble construction and evaluation
    3.1.      Model rating
        3.1.1.    Method
        3.1.2.    Application
    3.2.      Ensemble-mean performance
        3.2.1.    Based on model rating
        3.2.2.    Based on simulated non-informativeness
    3.3.      Probabilistic application
        3.3.1.    Skill and reliability
        3.3.2.    Loss estimation

*- How did the Authors select the models for their work? The paper by Gerl et al (2016) reviews a larger number of models, so the Authors need to explain the criteria applied for their selection.*

We agree that this is a relevant question, which requires addressing in the manuscript. From the paper by Gerl *et al.* (2016), we first pre-selected all deterministic flood vulnerability models describing loss to the asset type this work focuses on (residential buildings), and then excluded models based on following criteria:

- The documentation is insufficient for model implementation;
- The model uses explanatory variables that are not available in most practical applications;
- The model has a functional form that is judged inappropriate in the light of the state of the art in flood loss modelling (e.g. too simplistic or discretised);
- The model is based on the same underlying dataset of another model deemed more appropriate for the application settings (this is to ensure model independence and avoid potential biases in the ensembles).

We propose to add this explanation in Section 2.1.

*- An important point that I miss regards models availability. As a matter of fact, several flood damage models are hardly usable in practice, either because not accessible (e.g. commercial models), or because the publicly available information is incomplete and does not allow application (e.g. some research models). Are the models selected by the Authors freely accessible? This would be a major point to foster the use of multimodel ensembles as recommended by the Authors.*

The documentation of all models implemented in this study are openly accessible. References are provided in Table I, such that readers may consult the specific formulations of the models. We agree that this is an important point, which we will emphasize in the revised manuscript.

*Minor points*

*- title of Section 3.3 is not much informative for the reader, please change it.*

This will be addressed according to the reply to the first question.

*- Please indicate the measure unit in Tables 3 and 5.*

This will be added (Table 3: million €; Table 5: €).

*- Table should be numbered according to the order of citation in the text.*

This will be corrected.

*- Descriptions at page 11, lines 12-30 are not completely clear to me (e.g. I did not understand what measure is used to build the rank histogram), could you please add some more details?*

In the manuscript, we included this brief explanation to aid clarity, as readers from the flood loss modelling community may not be familiar with the Rank Histogram. However, because this is a widely used and well-documented method to assess the reliability of ensemble members, we think that describing this particular method in additional detail would be outside the scope of the manuscript. Two key references where additional information can be consulted are included.

---

## Author Comment (AC2) · 8 Mar 2018

**Response to Referee 2**

We would like to thank the referee for the time and effort put into reviewing the manuscript. This response carefully addresses all the comments. Where applicable, changes are proposed to the manuscript accordingly. Following the guidelines of the NHESS Editorial Board, the revised manuscript was not prepared at this point.

*General comments: - This paper is very interesting for the flood risk community.*

*- Title: The title could be more attractive. What accuracy will be improved? Accuracy of damage estimates?*

We agree, and propose the following new title:

Multi-model ensembles for assessment of flood losses and associated uncertainty

*- Abstract: The abstract could be formulated even more concrete and to the point. In addition, I miss here the view of the concrete result of the paper. The abstract is very general. That could make you more attractive.*

We agree, and propose the following changes accordingly (marked up for clarity):

Flood loss modelling is a crucial part of risk assessments. However, it is subject to large uncertainty that is often neglected. Most models available in the literature are deterministic, providing only single point estimates of flood loss, and large disparities tend to exist among them. Adopting any one such model in a risk assessment context is likely to lead to inaccurate loss estimates and sub-optimal decision-making. In this paper, we propose the use of multi-model ensembles to address these issues. This approach, which has been applied successfully in other scientific fields, is based on the combination of different model outputs with the aim of improving the skill and usefulness of predictions. We first propose a model rating framework to support ensemble construction, based on a probability tree of model properties, which establishes relative degrees of belief between candidate models. Using twenty flood loss models in two test cases, we then construct numerous multi-model ensembles, based both on the rating framework and on a stochastic method, differing in terms of participating members, ensemble size and model weights. We evaluate the performance of ensemble means, as well as their probabilistic skill and reliability. Our results demonstrate that well-designed multi-model ensembles represent a pragmatic approach to consistently obtain more accurate flood loss estimates and reliable probability distributions of model uncertainty.

*- What I miss completely is the explanation of how the 20 models were applied to the areas:*

*\* Which adjustments had to be made in the model or in the dataset (resolution, aggregation of land use)?*

*\* Which assumptions had to be made?*

*\* What land use data was used for Italy?*

*\* One should contrast the input variables for both cases in a table.*

*\* Can one compare both cases without hesitation or do the input data / assumptions differ too much?*

*I do not see this kind of discussion in the paper.*

- Section 2.2 provides a concise description of all relevant features of the hazard and exposure datasets used in this study. No adjustments are required in order to use these datasets for model application. The data they contain, shown in detail in Tables II and IV, are directly used as input to the vulnerability models, which establish a relation between those data and loss amounts.

- In cases where vulnerability models do not use some of the available hazard or exposure variables as explanatory variables, the unused variables are just not considered. For clarity, this will be mentioned explicitly in the manuscript. A description of how vulnerability models were applied is available at P4 L1-2, but we agree that this should be more informative. Accordingly, the sentence "In the estimation of losses for each asset, the best-matching function from each model is adopted." will be expanded in the revised manuscript.

- In the Italian case study, the exposure dataset contains spatially explicit residential buildings, which are directly used for the loss estimation. No additional land use data was necessary.

- Following the referee's suggestion, we will merge Tables II and IV.

- The hazard and exposure datasets contain slightly different information, as shown in Table II and IV (which will be merged). This has relevance essentially from the perspective of how certain vulnerability models are applied, for example when specific input data for certain vulnerability models are not available (e.g. what value to consider for flood velocity in the Mulde case, where only an indicator is available?). In the revised manuscript, we will include additional explanations on this matter.

*- There is no representation of the damage estimates of the models. How are the different locations considered in the results of the damage models? Why does one model give good results for Germany and bad ones for Italy? In my opinion, there is no detailed discussion of the damage estimates and the model results.*

The damage estimates given by each model are included as supplementary material (https://www.nat-hazards-earth-syst-sci-discuss.net/nhess-2017-349/nhess-2017-349-supplement.zip). It is not straightforward (or in most cases possible at all) to assess why one model gives good results for one location and bad results for another. This is part of the rationale behind the use of ensembles: if one could always know beforehand which model would perform best in a certain application, we could simply use that model instead. Therefore, we believe that a detailed discussion of the individual model performances is out of scope of this manuscript. We provide a detailed discussion of the ensemble performance in Section 3.2, which includes some discussion on individual models from P8 L34 to P9 L6. Aspects related with individual model performance are also addressed in the Introduction, which discusses issues such as model transferability, selection and uncertainty.

*Specific comments: - Please see the attached file.*

We agree with all the comments related with writing style, and in the revised manuscript will make corrections accordingly. Other specific comments are addressed below.

P3 L3: *You are using this wording (optimal decision-making) quite often. You should clearify what you mean by this.*

We propose to add the following sentence at P2 L31:

"Such estimates are unable to meet the decision needs of different stakeholders, who may have differing risk attitudes or cost-benefit ratios for precautionary measures (Merz and Thieken, 2009)."

P4 L31: *from whom?*

This is an official loss estimate provided in the report referenced in next sentence (Regione del Veneto, 2011b).

P5 L19: *Explain the data needed for the models (e.g. with a table), their boundary conditions and their uncertainty / complexity.*

In Table I, we have included all relevant information related with the use of the flood loss models in the scope of this study. This table explicitly shows the variables used by each model. Additional information may be consulted in the manuscript by Gerl *et al.* (2016), which is properly referenced in the manuscript. In P3 L28, we clearly state that all 20 models are deterministic (therefore, they do not provide uncertainty estimates).

P6 L34: *what are properties 1 and 2?*

These refer to Figure 3. This will be emphasized in this sentence to aid clarity.

P10 L16: *Please use formulas to explain the probabilistic method more clearly.*

Presumably, this comment refers to the Continuous Ranked Probability Score. The formula to calculate this evaluation metric will be added to the revised manuscript.

P11 L21: *What is the value of n?*

$n = 296$. We will add this information in the revised manuscript.

P13 L6-13: *In this section the discussion part is too short. Please rename the chapter into "discussion and conclusions" and discuss the pros and cons more comprehensively.*

The manuscript is structured such that all the results are discussed after they are presented, rather than in a separate section at the end. In particular, Section 3 follows a line of thought where the justification for each step of our study is made clearer after the previous step has been presented, applied, and discussed. We strongly believe that this logic for the paper structure aids readability, and therefore propose to keep it (with the improvements proposed

in the response to Referee 1). We will improve specific aspects of the discussion in Section 3, in particular related with the number of adopted models.

Figures 1 and 2: *Add a small map with the location of the catchment in Europe.*

This will be added in the revised manuscript.